# An Observational Cross-Sectional Analysis of the Correlation between Clinical Competencies and Clinical Reasoning among Italian Registered Nurses

**DOI:** 10.3390/healthcare12131357

**Published:** 2024-07-08

**Authors:** Ippolito Notarnicola, Blerina Duka, Marzia Lommi, Emanuela Prendi, Dhurata Ivziku, Gennaro Rocco, Alessandro Stievano

**Affiliations:** 1Centre of Excellence for Nursing Scholarship, OPI, 00146 Rome, Italy; genna.rocco@gmail.com (G.R.); alessandro.stievano@gmail.com (A.S.); 2Faculty of Medicine, University “Our Lady of the Good Counsel”, 1001 Tirana, Albania; bleriduka@yahoo.it (B.D.); e.prendi@unizkm.al (E.P.); 3Department of Clinical and Molecular Medicine, Faculty of Medicine and Psychology, University of Rome “La Sapienza”, 00157 Rome, Italy; marzia.lommi@uniroma1.it; 4Department of Health Professions, Fondazione Policlinico Universitario Campus Bio-Medico, 00128 Rome, Italy; d.ivziku@policlinicocampus.it

**Keywords:** clinical competence, clinic reasoning cross-sectional study, Nurse Competence Scale, nursing, nurse

## Abstract

Ability, knowledge, aptitude, and skill are the terms identified in the literature as the attributes of the concept of clinical competence. This implies that in order to act competently in their own context, the nurse must be able to make decisions which mainly depend on the ability to put clinical reasoning into practice. However, the evaluation of clinical reasoning in the various clinical-care activities of nursing competence is a necessary operation to prevent routine attitudes. From the perspective of an assessment of nursing competences, the aim of this study is to validate the relationship between the degree of competence recognized in a specific clinical setting and the amount of clinical reasoning executed by nurses. The study design was a cross-sectional observational design, following the guidelines of the Strengthening Reporting of Observational Studies in Epidemiology (STROBE) of observational studies. Both the Italian Nurse Competence Scale and the Nurse Clinical Reasoning Scale were used. The data was collected between 25 January and 5 March 2022. Four hundred twenty-four clinical nurses participated by completing and returning the questionnaires. The instruments underwent assessment to ensure internal consistency and test–retest reliability. Their validity was tested with the validity of known content, construct, and groups. This is supported by statistically significant correlations between the different variables examined and the scores of the different dimensions of the Italian Nurse Competence Scale and the Italian Nurse Clinical Reasoning Scale. The data collected showed an excellent average level of competencies and clinical reasoning, M = range of 72.24 and 63.93, respectively. In addition, we observed satisfactory scores across all dimensions of I-NCS (significance range: 0.000–0.014) and I-NCRS (significance range: 0.000–0.004). The understanding and development of clinical reasoning has also brought out new aspects that require further research. This study provides a fresh perspective on the correlation between clinical competences and clinical reasoning, representing a novel attempt to analyze their relationship.

## 1. Introduction

In today’s complex healthcare environments, nurses are pivotal in ensuring patients receive optimal care [1]. Critical to their effectiveness is the application of clinical reasoning, which integrates critical thinking, knowledge, and practical experience to navigate clinical challenges and make informed decisions [2,3].

According to Bussard et al., clinical reasoning is foundational to successful nursing practice [4], enabling nurses to achieve positive care outcomes through informed clinical judgments [5]. However, determining when newly graduated nurses attain full competence remains a significant challenge [6], exacerbated by reported deficiencies in clinical reasoning among novice nurses [1,7], which are crucial for their professional integration [8].

To bridge the theory–practice gap, it is essential to evaluate clinical reasoning across various nursing activities for novice nurses and encourage experienced professionals to adopt innovative approaches [2,9]. Patient safety hinges on nurses’ competence, ensuring adherence to care standards and the delivery of high-quality care [10,11].

Despite ongoing debate, nursing competence is broadly defined as the ability to perform tasks that lead to desired outcomes in real-world scenarios [12]. Meretoja et al. advocate for a holistic understanding of competence, emphasizing the integration of knowledge, ability, skills, attitudes, and values specific to clinical settings to effectively meet care standards [13,14].

The literature identifies competence in nursing as encompassing abilities, knowledge, aptitude, and skills [14], aligned with Benner’s stages of competence acquisition, which progress from novice to expert through experiential learning [15].

For instance, a level of greater experience that enables one to construct a more comprehensive holistic perspective is contrasted with a competent level of performance that permits operating safely. In particular, a competent level of performance is distinguished, which allows you to act safely, as distinct from a level of greater experience, which allows you to develop a broader holistic concept. Benner equates the level of experienced nurses to experience gained in a clinical setting for two or three years [15]. The nursing competency assessment is a procedure that has an impact on healthcare organizations when they are obliged to gauge nurses’ levels of competency in order to raise the standard of care [16]. Regarding the importance of assessing competences, it is essential to consider how they are measured and the tools that must be used in the evaluation process [17].

Several tools have been developed for this purpose, such as the Nurse Competence Scale, which was developed by Meretoja et al. between 1997 and 2004 and is used for the self-assessment of nursing competencies. It is based on the competence model developed by Benner [14,18,19,20]. The technique has been used in numerous other nations, demonstrating its validity and dependability.

They made several modifications to the German version of the Nurse Competence Scale in a psychometric validation study, confirming a scale with six categories and fifty-four items [21]. In contrast, the Norwegian version approved forty-six items arranged into four categories, compared to the original instrument [22]. 

While Finotto and Cantarelli translated and validated the Italian version of the Nurse Competence Scale in 2009 [23], Notarnicola et al. developed the Italian Nurse Competence Scale (I-NCS) and assessed the psychometric properties of the scale by testing its construct validity, confirming the scale’s seven original categories while reducing the number of items to fifty-eight [24].

Clinical reasoning is a complex process where nurses apply critical thinking skills with a patient-centered focus, utilizing cognition, metacognition, and a professional knowledge base to gather and analyze patient data, and implement appropriate actions in clinical settings [7]. Benner et al. [25] highlight differences in communication styles and decision-making modalities between experienced nurses and novices, emphasizing the role of clinical reasoning shaped by professional experience [25].

Benner et al. define clinical reasoning as “decision-making based on reasoning within a clinical context,” considering it a core skill for nurses [25]. They suggest that action-oriented questions are crucial in applying clinical reasoning effectively, especially in uncertain clinical scenarios tailored to individual patient needs [26].

Recent definitions of clinical reasoning underscore its role as a complex cognitive process involving formal and informal thinking strategies to collect, analyze, and interpret patient information, and to evaluate alternative actions [27]. This process is essential for nurses to identify, evaluate, and address patient care issues, making informed clinical decisions and assessing outcomes accordingly [5].

Recent research explores clinical reasoning and decision-making as multifaceted processes influenced by social, psychological, cultural, contextual, emotional, and cognitive factors [28,29]. Nurses lacking adequate clinical reasoning skills may struggle to recognize and address patient care needs effectively [7].

Despite its recognized importance in nursing competence, the development of a clinical reasoning rating scale is relatively recent [30]. The Nurse Clinical Reasoning Scale (NCRS), based on a clinical reasoning model, is a validated tool developed by Liou et al. to assess nurses’ clinical reasoning skills reliably [30]. This scale, depicted as an eight-step cyclical process by Levett-Jones et al., includes observation, identification, processing, decision-making, planning, action, evaluation, and reflection [31].

Initially in Chinese, the NCRS has been adapted and evaluated for use in different contexts, such as the Italian Nurse Competence Reasoning Scale (I-NCRS) developed by Notarnicola et al. This adaptation process enhances the understanding of nursing competencies in diverse cultural and professional settings, maintaining the scale’s validity and reliability [32].

At a global level, clinical competencies and clinical reasoning play a crucial role in ensuring the quality and effectiveness of nursing care, directly influencing healthcare practices worldwide. The study of such issues in Italy holds particular significance, allowing for a comprehensive understanding of the challenges and peculiarities of the national healthcare system and thus contributing to enhancing nursing practices both locally and globally.

In the field of competence research, our study represents a fundamental contribution that introduces new methodological and theoretical perspectives. This seminal work aims to redefine and expand the understanding of clinical competencies, providing a solid foundation for future research and practical applications.

The purpose of this study is to confirm the relationship between the level of competence perceived in a particular clinical environment and the extent of clinical reasoning performed by nurses from the standpoint of an assessment of nursing competencies. The specific goals of our study were to assess clinical reasoning using the Italian version of the I-NCRS; assess clinical competence among registered nurses in clinical practice using the Italian version of the I-NCS; and look into possible relationships between the development of clinical competencies and clinical reasoning among Italian nurses.

## 2. Materials and Methods

### 2.1. Study Design

The design of the study was a cross-sectional design. The description of the study was carried out following the guidelines of the Strengthening Reporting of Observational Studies in Epidemiology (STROBE) of observational studies [33].

The number of samples was calculated by power analysis. The sample was calculated in effect size of 0.16, theoretical power of 80%, and confidence level of 95%, and the number of samples was determined as 414 [34]. The sample was made up of registered nurses with at least 5 months of work in both clinical and territorial settings who agreed to participate in the research.

We conducted a cross-sectional survey across multiple Italian hospitals (Northern, Central, and Southern Italy), selecting a convenience sample of around 424 registered nurses employed in public hospital settings. Data were collected between 25 January and 5 March 2022, and a total of 500 questionnaires were distributed to the participating hospitals.

The inclusion criteria were that all participants should be registered nurses with over 5 months of experience in clinical practice in different settings (Table 1). Each participant received two questionnaires, along with a consent form and two response envelopes bearing their address. Data collection utilized anonymous, self-administered, and structured questionnaires. Upon completion, participants sealed and returned the questionnaires and consent forms in the provided envelopes, maintaining confidentiality throughout. To ensure data privacy, all completed questionnaires were handled without identifiable tags or specific personal information, and our study data kept by a researcher (IN). Excluded from the sample were student nurses and non-registered nurses, and registered nurses who had less than 5 months of clinical practice. 

### 2.2. Instrument

For the assessment of clinical competences, the Italian version of the Nurse Competence Scale (I-NCS) was used in our study [30]. The I-NCS, a self-assessment tool for the perception of clinical nursing competencies, is divided into 7 dimensions, derived from the framework of competencies defined by Patricia Benner [35]. The dimensions of the I-NCS include a series of specific competencies which are: management skills (7 items); educational competencies (16 items); diagnostic skills (7 items); situation management skills (8 items); therapeutic interventions (10 items); ensure quality (6 items); role held (19 items). The total number of items that make up the I-NCS is 73. Proficiency level is measured through a Visual Analog Scale (VAS), where zero indicates a very low level of proficiency and 100 indicates a high level of proficiency. The VAS scale, for descriptive purposes only, has been divided into 4 ranges to express with a judgment the level of competence of the nurses, starting from poor (0–25), rather good (>25–50), good (>50–75), and very good (>75–100). 

As far as the evaluation of clinical reasoning is concerned, it was carried out with the administration of the Italian version of the Nurse Clinical Reasoning Scale (I-NCRS) instrument [31]. The NCRS was developed in 2015 by Liou and colleagues [30] at Taiwan University on the conceptual model developed by Levett-Jones [31]. The I-NCRS, a self-assessment tool for the perception of clinical reasoning, consists of 15 items rated on a 5-point Likert scale (1 = strongly disagree, 2 = disagree, 3 = neutral, 4 = agree, 5 = strongly agree). The sum of the total item scores for the tool ranges from 15 to 75; starting from poor (15–35), good (36–55), and very good (56–75). A higher score indicates a higher level of clinical reasoning ability. The instrument in question demonstrated high reliability in the original study (Cronbach’s Alpha = 0.94) [29]. The NRCS has been validated and translated into several languages and has been adapted to the Italian context by Notarnicola and colleagues, demonstrating high reliability (Cronbach’s Alpha = 0.90) and stability (ICC = 0.90; CI = 0.87–0.92) [33].

### 2.3. Analytical Statistics

The data analysis was conducted using SPSS for Windows, version 26 (SPSS Inc., Chicago, IL, USA).

The answers for each competence area were converted into scores between 1 and 100, where 100 represented the highest competency and 1 the lowest, in keeping with the instructions provided in the I-NCS user manual. In a similar vein, the answers for every clinical reasoning domain were recalculated using a scale that went from 15 to 75, where 75 denoted advanced clinical reasoning and 15 the lowest level.

Calculations were made for descriptive statistics, including means and standard deviations (SD), based on the provided demographic data, with frequencies and percentages. For three and more group comparisons, inferential statistics, such as one-way analysis of variance (ANOVA), were used. The correlations between the I-NCS factor scores were examined using Pearson’s correlation coefficient in order to assess the adequacy of work competencies in a professional context. Using Pearson’s correlation, continuous data were correlated. A *p*-value of less than 0.05 was deemed statistically significant. [36]. The internal consistency of each area of expertise and clinical reasoning was calculated using Cronbach’s alpha. The computed average of the absent data points has been used to impute the missing values. Cases that lacked at least 50% of the replies were not included in the analysis. Three of the writers separately carried out the data analyses (I.N., A.S. and G.R.).

### 2.4. Ethical Considerations

The sample for this study was made up of registered nurses, and no patients were included. Standard Operating Procedures (SOPs) and Good Clinical Practice (GCP) provide an explanation of international scientific and ethical quality standards that were followed in the design, conduct, registration, and reporting of the study. During the data collection and analysis procedures, participants were additionally informed of the confidentiality and anonymity of their responses. The Center of Excellence for Nursing Scholarship’s (CECRI) Ethics Committee granted ethical approval for this project (21 February 2022).

## 3. Results

### Data Collection

Approximately 500 I-NCS questionnaires were distributed, and the response rate was 84.8% (*n* = 424). All nurses who returned the questionnaire with at least 80% of the answers completed correctly were included in the present study. Women made up the majority (64.6%). Table 1 showcases comprehensive demographic data.

Cronbach’s alpha values for the seven I-NCS factors ranged from 0.899 to 0.919 and the value was 0.920 for the total scale, while the three factors of the IRCS scale ranged from 0.736 to 0.855 and the value was 0.857 for the total scale (Table 2).

Table 3 shows a statistically significant positive correlation between Role (Professional Awareness (F = 8.615; *p* value 0.000), Tutorial Functions (F = 5.319; *p* value 0.005*), and Professional Leadership (F = 7.317; *p* value 0.001*)) of the I-NCS scale. Among Age (Tutorial Functions (F = 5.264; *p* value 0.001)) of the I-NCS scale. Among Sex (Tutorial Functions (F = 4.573; *p* value 0.033)) of the I-NCS scale; and finally, between Job Satisfaction (Nursing Research F = 5.398; *p* value 0.001), Professional Awareness (F = 6.812; *p* value 0.000), Ethical Value (F = 11.299; *p* value 0.000), Tutorial Functions (F = 3.592; *p* value 0.014), Professional Leadership (F = 6.388; *p* value 0.000), Educational Interventions (F = 5.176; *p* value 0.002), and Management for Care Process (F = 8.053; *p* value 0.000) showed a statistically significant positive correlation with all dimensions of the I-NCS scale.

Table 4 shows a statistically significant positive correlation between Operating Unit (Nursing Health Information (F = 2.153; *p* value 0.037*)) of the I-NCRS scale; and between Job Satisfaction (Nursing Health Problems (F = 6.243; *p* value 0.000*) and Nursing Health Information (F = 6.784; *p* value 0.000*). Nursing Health Assessment (F = 4.512; *p* value 0.004*), showed a statistically significant positive correlation with all dimensions of the I-NCRS scale. 

## 4. Discussion

Clinical competence is an important aspect of nursing education, clinical environments, and nurse administration. It is the capacity to integrate knowledge, skills, attitudes, and abilities into a clinical setting. In nursing, problem-solving and decision-making are based on clinical reasoning. By assisting in the identification of the patient’s needs, facilitating efficient treatment planning, and enhancing results, it enhances patient care. It is essential to the security and caliber of medical care.

This study aimed to analyze the clinical competences and clinical reasoning level of registered nurses enrolled in the investigation using the I-NCS and I-NCRS. Therefore, the data collected showed an excellent average level of competencies and clinical reasoning (M = range of 72.24 and 63.93, respectively). 

Our study’s findings indicate that the clinical reasoning and clinical competencies of the nurses in our sample seem to evolve along a competence development trajectory, supported by statistically significant correlations between the variables examined and the scores across dimensions of the I-NCS and I-NCRS scales (Table 3 and Table 4).

In Table 3 and Table 4, as we can see, one of the factors under analysis that has a statistically significant positive influence on the growth of clinical competence and clinical reasoning is job satisfaction; in fact, this can be seen in Table 3 and Table 4 where the statistical significance is more evident in all dimensions of the scales used. The same can be observed for Role, Age, and Sex with I-NCS; and Operating Unit with I-RCS.

Indeed, the variable of job satisfaction has been observed in various international studies and has the potential to impact competence development, effective nursing clinical reasoning, and nursing care. According to a study conducted by Zakeri et al. on the relationship between job satisfaction and nursing clinical competences, it was found that nursing professionals with high levels of job satisfaction also demonstrate greater clinical competence, confirming the results found in our survey [37]. In our research, we observed satisfactory scores in all dimensions of I-NCS, with a significance range between 0.000 and 0.014. This is crucial because effective clinical practice necessitates nurses with strong clinical competencies and a high degree of job satisfaction based on these variables [38]. Also, all dimensions of I-NCRS obtained positive scores in our study, with a significance range between 0.000 and 0.004. This implies that job satisfaction contributes to improving decision-making in the clinical setting, promoting better management of the quality of care provided by nurses. A study conducted by Niskala et al. has also highlighted how job satisfaction directly influences the quality of nursing care, further reinforcing the results from our research [39]. Our study has shown that nurses are satisfied with their work and consider it essential. This is in line with international literature that has highlighted how professionals with a strong professional identity provide valid patient care in healthcare settings and develop better clinical competences to increase patient satisfaction.

Our study also sought to determine whether clinical reasoning and the development of clinical competences were correlated. From our analysis, it emerges that all dimensions of I-NCS are not correlated with I-NCRS (see Table 5). The concept that clinical reasoning has no impact on the development of clinical competences is important, as extensive research indicates that clinical reasoning is central to the integration of theoretical knowledge into clinical practice and the ability to deal with complex situations effectively [40]. Clinical reasoning allows nurses to analyze patient data, make accurate diagnoses, plan appropriate interventions, and continually adapt their practice based on emerging evidence. Therefore, it is important that the nursing process is taught completely to improve clinical reasoning as well. Without sound clinical reasoning, nurses may not be able to make informed and timely decisions, thus compromising the quality of care provided to patients.

Our findings indicate a positive correlation between job satisfaction and multiple dimensions of the I-NCRS and I-NCS scales. The writers’ goal of having the broadest coverage of the subject has been achieved. Individual differences in qualities, attributes, and communication abilities give rise to multiple components of clinical reasoning and clinical competences. Therefore, it is essential for a nurse to know when to display a specific skill, ability, attitude, and understanding of the fundamentals of competences in clinical practice [41]. This also relies on the self-efficacy of each nurse, having the perception of their own ability to professionally perform their nursing duties.

Moreover, our findings indicate that additional research is necessary to determine which other factors may affect nursing proficiency. This highlights the need for additional studies that cover a broad range and help shed light on an area that is still primarily empirical and lacking in scientific data. The fact that our study produced positive job satisfaction outcomes shows that various cultures, contexts, and people call for various behaviors and approaches that change over time based on the circumstances. 

Due to professional competencies and a positive cultural attitude, job satisfaction plays a crucial role in fostering healthy relationships for patient safety. In order to assist nurses to stay optimistic throughout their careers, nursing school should place equal emphasis on fostering a pleasant work culture and supplying them with the necessary professional competences. It is crucial for an organization to have a culture of good job satisfaction when new nurses join. It is highly recommended that educators and nursing managers not only give professional education for nurses, but also offer sufficient training to enable them to identify and enhance their job happiness, since a positive work environment leads to higher levels of personal satisfaction. Positive job satisfaction among nurses enables them to create better standards and operational procedures, which in turn provides patients with the highest caliber of treatment. In the end, nurses’ views regarding patient safety culture and the caliber of care given are influenced by their job happiness. Future research on the advantages of job happiness for nurses is warranted by our findings.

### Limitations

The limitations of a correlational study are evident in our study. The cross-sectional study design made it impossible to identify causal relationships between the variables examined. Nursing competence attitude, patient advocacy, and quality of care and its predictors should be assumed with caution. The sample tested was also cost-effective. As a result, we are unable to generalize the data; future research should aim to improve these elements. However, these findings make further research on job satisfaction in other areas of nursing work necessary. Furthermore, our study shows that job satisfaction in relation to clinical competences and clinical reasoning is statistically significant; future investigations should be aimed at finding possible predictors. Despite these limitations, this study fills an important gap in the current scientific literature by examining the relationship between clinical competences and clinical reasoning among nursing staff, a relatively under-explored area in the literature. However, it is important to underline that, within this study, the lack of relationship between the two variables was confirmed. This study provides a fresh perspective on the correlation between clinical competences and clinical reasoning, representing a novel attempt to analyze their relationship.

## 5. Conclusions

In conclusion, the results of our study provide a deeper and more articulated understanding of clinical competencies. This seminal work not only lays the groundwork for further investigations but also has the potential to significantly influence educational and clinical practices, promoting a more integrated and comprehensive approach to competence development. The results of our analysis demonstrated that clinical reasoning has no impact on the development of clinical competences. In fact, no statistical significance was found in the correlation between the I-NCS and I-NCRS scales. This is probably due to the fact that some authors state that clinical reasoning concerns the ability of a nurse to use their clinical knowledge, understand the culture of the context in which they work and use the principles of medical ethics to accurately assess the condition of a patient and choose the most appropriate treatment for them. Therefore, it has no impact on the actual level of competence. However, the results of our investigation demonstrated that job satisfaction is closely related to the development of clinical competences and clinical reasoning.

## Figures and Tables

**Table 1 healthcare-12-01357-t001:** Sociodemographic data (*n* = 424).

		*n*	%
Role	Nurse	290	68.4
	Nurse Coordinator	111	26.2
	Nurse Manager	23	5.4
Sex	F	274	64.6
	M	150	35.4
Age	21–30	73	17.2
	31–40	69	16.3
	41–50	139	32.8
	51+	143	33.7
Operating Unit	Ambulatory	20	4.7
	Radiodiagnostics Area	3	0.7
	Surgical Area	62	14.6
	Critical Area	98	23.1
	Training Area	20	4.7
	Medical Area	130	30.7
	Territorial Area	75	17.7
	Health Management	16	3.8
Current Operating Unit years	1–10	279	65.8
	11–20	100	23.6
	21–30	41	9.7
	31–40	4	0.9
Exercise profession for years	1–10	114	26.9
	11–20	97	22.9
	21–30	119	28.1
	31–40	91	21.5
	41–50	3	0.7
Job Satisfaction	Moderate	213	50.2
	High	167	39.4
	None	6	1.4
	Low	38	9.0

**Table 2 healthcare-12-01357-t002:** Cronbach’s Alpha of the I-NCS and I-NCRS scales.

Scale	Dimension	Mean	SD	Alpha di Cronbach	Total Alpha di Cronbach
I-NCS	Using Research	72.24	13.26	0.908	0.92
	Professional Awareness	81.65	11.11	0.913	
	Ethical Value	75.78	12.74	0.901	
	Tutorial Functions	78.08	15.97	0.919	
	Professional Leadership	78.39	11.88	0.899	
	Educational Interventions	75.18	14.54	0.914	
	Management for Care Process	80.87	11.38	0.905	
	Total	77.46			
I-NCRS	Nursing Health Problems	63.93	7.75	0.812	0.857
	Nursing Health Information	63.20	8.69	0.736	
	Nursing Health Assessment	61.81	10.37	0.855	
	Total	62.98			

**Table 3 healthcare-12-01357-t003:** Association between socio-demographic data and I-NCS scale.

I-NCS								
		Nursing Research	Professional Awareness	Ethical Value	Tutorial Functions	Professional Leadership	Educational Interventions	Management for Care Process
Role	F	2.705	8.615	2.662	5.319	7.317	0.104	1.526
	Sign.	0.068	0.000 *	0.071	0.005 *	0.001 *	0.902	0.219
Operating Unit	F	1.608	1.334	1.727	0.985	1.102	3.072	2.044
	Sign.	0.131	0.232	0.101	0.442	0.361	0.004 *	0.049 *
Age	F	1.143	0.45	0.255	5.264	2.493	0.917	0.798
	Sign.	0.331	0.717	0.858	0.001 *	0.06	0.433	0.495
Sex	F	2.041	0.005	0.022	4.573	0.507	1.946	0.35
	Sign.	0.154	0.944	0.883	0.033 *	0.477	0.164	0.555
Exercise profession for years	F	0.369	0.989	0.258	1.944	1.324	0.789	0.291
	Sign.	0.83	0.413	0.905	0.102	0.26	0.533	0.884
Current Operating Unit years	F	0.297	0.247	0.575	1.247	1.373	0.15	0.29
	Sign.	0.828	0.864	0.632	0.292	0.25	0.93	0.833
Job Satisfaction	F	5.398	6.812	11.299	3.592	6.388	5.176	8.053
	Sign.	0.001 *	0.000 *	0.000 *	0.014 *	0.000 *	0.002 *	0.000 *

* Significant values.

**Table 4 healthcare-12-01357-t004:** Association between socio-demographic data and I-NCRS scale.

I-NCRS				
		Nursing Health Problems	Nursing Health Information	Nursing Health Assessment
Role	F	0.373	0.424	0.078
	Sign.	0.689	0.655	0.925
Operating Unit	F	1.703	2.153	1.217
	Sign.	0.107	0.037 *	0.292
Age	F	1.817	0.809	1.877
	Sign.	0.143	0.490	0.133
Sex	F	0.665	0.822	0.009
	Sign.	0.415	0.365	0.923
Exercise profession for years	F	0.622	0.756	1.489
	Sign.	0.647	0.554	0.205
Current Operating Unit years	F	0.548	0.218	1.621
	Sign.	0.649	0.884	0.184
Job Satisfaction	F	6.243	6.784	4.512
	Sign.	0.000 *	0.000 *	0.004 *

* Significant values.

**Table 5 healthcare-12-01357-t005:** Correlation between I-NCS and I-NCRS scale.

		I-NCS							I-NCRS		
		Using Research	Professional Awareness	Ethical Value	Tutorial Functions	Professional Leadership	Educational Interventions	Management for Care Process	Nursing Health Problems	Nursing Health Information	Nursing Health Assessment
I-NCS	Using Research	1									
	Professional Awareness	0.634 **	1								
	Ethical Value	0.731 **	0.627 **	1							
	Tutorial Functions	0.569 **	0.542 **	0.608 **	1						
	Professional Leadership	0.696 **	0.650 **	0.754 **	0.674 **	1					
	Educational Interventions	0.573 **	0.493 **	0.677 **	0.531 **	0.669 **	1				
	Management for Care Process	0.631 **	0.659 **	0.698 **	0.579 **	0.784 **	0.671 **	1			
I-NCRS	Nursing Health Problems	0.267 **	0.268 **	0.308 **	0.222 **	0.298 **	0.319 **	0.332 **	1		
	Nursing Health Information	0.222 **	0.246 **	0.305 **	0.187 **	0.319 **	0.341 **	0.313 **	0.752 **	1	
	Nursing Health Assessment	0.215 **	0.170 **	0.254 **	0.262 **	0.289 **	0.340 **	0.274 **	0.607 **	0.695 **	1

**. The correlation is significant at the 0.01 level (2-tailed).

## Data Availability

The data presented in this study are available within the article.

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
