# Peer review of "An Observational Cross-Sectional Analysis of the Correlation between Clinical Competencies and Clinical Reasoning among Italian Registered Nurses"

_healthcare, 2024, doi:10.3390/healthcare12131357_

Round 1
Reviewer 1 Report
Comments and Suggestions for Authors
Comments on Manuscript:
Line 12 the word ability appears twice.
References all quite old, 10 years or more
In America, the AACNs competency-based education and the new AACN Esentials 2021 have generated a great deal of literature in the last 4 years about competency and competency evaluation.
Competency definitions using references 22 years old Meretoja et al (2002).
Line 26 Our study’s findings demonstrate that the clinical reasoning and clinical competences of the sample of nurses under investigation appear to follow a competence development trajectory that is bolstered by statistically significant correlations between the va ious variables under investigation and the partial scores of the dimensions of the INCS and IRCS scales (Tables 3 and 4). (What does this sentence mean?) Table 3 is labeled sociodemographic data and the IRCS. I thought it was competencies under investigation?
Line 67 Skills knowledge attitudes and skills The word skills twice
Line 75 begins with In instance and I wonder if it should be For instance? Also in row 75, the reference to “you” makes me wonder if this portion of the text was written by AI generation
Line 77 “According to Patricia Benner (1984), experience earned in a clinical context for two or three years is equivalent to the degree of experience had by a nurse” I’m not sure what this sentence is trying to say. Are nursing programs 2 or 3 years long and do they provide experience? I hope so but not sure what meaning this sentence is trying to provide.
Line 81: Regarding the significance of skill evaluation, it is critical to keep in mind the analysis of how skills are measured and the resources that should be employed [16]. This is a very general vague statement that perhaps will be fleshed out but is at this point left to stand on its own Another mark of AI generated sentences.
Line 89 Who is “He” ? Does the author mean Meretoja? That might be a logical conclusion from the previous sentence but I think Ritta Meretoja is a Finnish woman.
Line 101 “Recent studies” are not 14 years old, even if Benner’s work was classic.
Line 260 “Our study’s findings demonstrate that the clinical reasoning and clinical competences of the sample of nurses under investigation appear to follow a competence development trajectory that is bolstered by statistically significant correlations...”
The study does not demonstrate a relationship between clinical reasoning and clinical competency.
Line 271 Nursing cart is likely nursing care.
Lines 288-292. I’ll have to review Table 5 as it was not in this manuscript but this statement does not have other support or backup. This is also a self-administered scale not an observational analysis of competencies.
Line 332 I do not think the article identified a gap in the literature related to any relationship between clinical reasoning and clinical competence as much as it made a statement about their lack of relationship without references or explanation of their findings.
Line 336 Conclusions:
“The results of our analysis demonstrated that clinical reasoning has no impact on the development of clinical competencies. (This study data shows nothing of the kind.) This is probably due to the fact that some authors state that clinical reasoning concerns the ability of nurses to use their clinical knowledge, understand the culture of the context in which they work, and use the principles of medical ethics to accurately assess the condition of a patient and choose the most appropriate treatment for him. therefore, it has no impact on the actual level of competence. However, the results of our investigation demonstrated that job satisfaction is closely related to the de-elopement of clinical competencies and clinical reasoning.”Nonsensical conclusions.
Interesting that an author cited, Meretoja has a recent publication following these recommentations in the last sentence of the conclusion: Numminen O, Ruoppa E, Leino-Kilpi H, Isoaho H, Hupli M, Meretoja R. Practice environment and its association with professional competence and work-related factors: perception of newly graduated nurses. J Nurs Manag. 2016 Jan;24(1):E1-E11. doi: 10.1111/jonm.12280. Epub 2015 Feb 11. PMID: 25676482.
Comments on the Quality of English LanguageEnglish looks like much of it is AI-generated. I found only a few actual grammatical mistakes which I've listed above. Vague statements, old references and citations, and the use of the word "you" about the reader or nonsensical conclusions are hallmarks.
Author Response
REVIEWER REPLIES 1
Line 12 the word ability appears twice.
I thank the reviewer for the suggestion we replaced the word with skill, it's a typo.
References all quite old, 10 years or more
We thank the reviewer for the suggestion we have added the recent references.
In America, the AACNs competency-based education and the new AACN Esentials 2021 have generated a great deal of literature in the last 4 years about competency and competency evaluation.
Competency definitions using references 22 years old Meretoja et al (2002).
We thank the reviewer for the suggestion given, we used that reference as the I-NCS scale is based on the Meretoja definition and the same is based on the Benner definition. This is because ours is a seminal work to understand clinical skills.
Line 26 Our study’s findings demonstrate that the clinical reasoning and clinical competences of the sample of nurses under investigation appear to follow a competence development trajectory that is bolstered by statistically significant correlations between the va ious variables under investigation and the partial scores of the dimensions of the INCS and IRCS scales (Tables 3 and 4). (What does this sentence mean?) Table 3 is labeled sociodemographic data and the IRCS. I thought it was competencies under investigation?
We thank the auditor for the suggestion, we have changed tables 3 and 4 according to the advice of reviewer 4 as well, so we have improved the tables now the reader has a clearer idea and we stated that are competencies under investigations
Line 67 Skills knowledge attitudes and skills The word skills twice
I thank the reviewer for the suggestion we replaced the word with ability, it's a typo. It was a typos.
Line 75 begins with In instance and I wonder if it should be For instance? Also in row 75, the reference to “you” makes me wonder if this portion of the text was written by AI generation
We thank the reviewer for the suggestion given, we have changed "in" to "For", we have also changed the sentence, in this way it is clearer and more readable, as in the other mode it could seem to have written the sentence with an AI.
Line 77 “According to Patricia Benner (1984), experience earned in a clinical context for two or three years is equivalent to the degree of experience had by a nurse” I’m not sure what this sentence is trying to say. Are nursing programs 2 or 3 years long and do they provide experience? I hope so but not sure what meaning this sentence is trying to provide.
We thank the reviewer for the suggestion given, we have changed the sentence, in this way it is more understandable to the reader of the magazine.
Line 81: Regarding the significance of skill evaluation, it is critical to keep in mind the analysis of how skills are measured and the resources that should be employed [16]. This is a very general vague statement that perhaps will be fleshed out but is at this point left to stand on its own Another mark of AI generated sentences.
We thank the reviewer for the suggestion given, we changed the sentence, in this way it is clearer and more readable as in the other mode it could seem to be written with AI.
Line 89 Who is “He” ? Does the author mean Meretoja? That might be a logical conclusion from the previous sentence but I think Ritta Meretoja is a Finnish woman.
We thank the reviewer for the suggestion, there was a typo and the exact word was "They", as the reference was the german changes is clearer.
Line 101 “Recent studies” are not 14 years old, even if Benner’s work was classic.
We thank the reviewer for the suggestion given, we have removed the word "recent", as it could give a wrong interpretation.
Line 260 “Our study’s findings demonstrate that the clinical reasoning and clinical competences of the sample of nurses under investigation appear to follow a competence development trajectory that is bolstered by statistically significant correlations...”
The study does not demonstrate a relationship between clinical reasoning and clinical competency.
We thank the reviewer for the suggestion given, we have changed the sentence to make it more understandable
Line 271 Nursing cart is likely nursing care.
We thank the reviewer for the suggestion given, we changed the word to "care"
Lines 288-292. I’ll have to review Table 5 as it was not in this manuscript but this statement does not have other support or backup. This is also a self-administered scale not an observational analysis of competencies.
We thank the reviewer for the suggestion given, we have inserted the missing table 5.
Line 332 I do not think the article identified a gap in the literature related to any relationship between clinical reasoning and clinical competence as much as it made a statement about their lack of relationship without references or explanation of their findings.
We thank the reviewer for the suggestion given, we have changed the sentence improving the comprehensibility making it clearer to the reader.
Line 336 Conclusions:
“The results of our analysis demonstrated that clinical reasoning has no impact on the development of clinical competencies. (This study data shows nothing of the kind.) This is probably due to the fact that some authors state that clinical reasoning concerns the ability of nurses to use their clinical knowledge, understand the culture of the context in which they work, and use the principles of medical ethics to accurately assess the condition of a patient and choose the most appropriate treatment for him. therefore, it has no impact on the actual level of competence. However, the results of our investigation demonstrated that job satisfaction is closely related to the de-elopement of clinical competencies and clinical reasoning.”Nonsensical conclusions.
We thank the reviewer for the suggestion given: we have changed the conclusions and improved them also based on the other suggestions of the reviewers. Now the conclusions are more comprehensible.
Interesting that an author cited, Meretoja has a recent publication following these recommentations in the last sentence of the conclusion: Numminen O, Ruoppa E, Leino-Kilpi H, Isoaho H, Hupli M, Meretoja R. Practice environment and its association with professional competence and work-related factors: perception of newly graduated nurses. J Nurs Manag. 2016 Jan;24(1):E1-E11. doi: 10.1111/jonm.12280. Epub 2015 Feb 11. PMID: 25676482.
Comments on the Quality of English Language
English looks like much of it is AI-generated. I found only a few actual grammatical mistakes which I've listed above. Vague statements, old references and citations, and the use of the word "you" about the reader or nonsensical conclusions are hallmarks.
We thank the reviewer for the suggestion given, the document has been completely revised in the quality of the English language making the manuscript clearer and more readable as in the other mode it could seem to be written with AI.
Reviewer 2 Report
Comments and Suggestions for Authors
1. The objective presented in the abstract is not the same as that stated at the end of the Introduction. Furthermore, at the end of the introduction the purpose is to confirm the relationship between variables, which is not justified as a purpose but, eventually, as a specific objective.
2. Introduction is extensive, although little space has been dedicated to referring to clinical reasoning models and justifying the choice of I-NCRS.
3. Study design, material and methods are well explained. There are no other studies with the same scale, although the reliability and stability of the scale are mentioned.
4. The results are presented in detail (long table) - probably, could be improved the presentation style.
5. The discussion is descriptive and comparative with few studies - however, the suggestions focus on the need for additional studies. No relationship with the nursing process or the use of clinical evidence was proposed.
6. The interpretative “leap” that clinical reasoning has no impact on the development of clinical skills should be more substantiated and justified.
Author Response
REVIEWER RESPONSES 2
- The objective presented in the abstract is not the same as that stated at the end of the Introduction. Furthermore, at the end of the introduction the purpose is to confirm the relationship between variables, which is not justified as a purpose but, eventually, as a specific objective.
- We thank the reviewer for his suggestion, we have aligned the objective at the end of the introduction with the objective of the abstract. Besides, we stated that the relationship between variables is a specific objective.
- Introduction is extensive, although little space has been dedicated to referring to clinical reasoning models and justifying the choice of I-NCRS.
- We modified the introduction and shortened it.
- Study design, material and methods are well explained. There are no other studies with the same scale, although the reliability and stability of the scale are mentioned.
- There is only one Persian study Hosseinzadeh, T., Mirfarhadi, N., Pouralizadeh, M., Tabrizi, K. N., Fallahi-Khoshknab, M., Khankeh, H. R., & Shokooh, F. (2024). Psychometric properties of the persian version of the nursing clinical reasoning scale. Nursing open, 11(1), e2041. https://doi-org.ezproxy.uniroma1.it/10.1002/nop2.2041
- The results are presented in detail (long table) - probably, could be improved the presentation style.
- Ringraziamo il revisore per il suo suggerimento, abbiamo modificato le tabelle in base al revisore 4 adesso sono migliorate nella leggibilità.
- The discussion is descriptive and comparative with few studies - however, the suggestions focus on the need for additional studies. No relationship with the nursing process or the use of clinical evidence was proposed.
- We thank the reviewer for the suggestion, we have added an additional sentence in the discussion.
- The interpretative “leap” that clinical reasoning has no impact on the development of clinical skills should be more substantiated and justified.
- We thank the reviewer for his suggestion, we have included a sentence that clarifies our interpretation.
Reviewer 3 Report
Comments and Suggestions for Authors
An Observational Cross-Sectional Analysis on the Correlation between Clinical Competencies and Clinical Reasoning among Italian Registered Nurses.
Title: No information about the location of the study. Study location information will help the reader identify the study easily.
Abstract
1. There is a duplicate word that is "ability" in the early abstract section. Please re-check the overall sentence in the manuscript to avoid typos.
2. Please emphasize the novelty of the study in the abstract section.
3. What does “The data was collected from 2022.” mean? Would it have the end of the data collection period? Please make it clear.
4. Please add the statistical values of the study result.
Introduction
1. There are some inconsistent reference styles (numbered and by name and year).
2. In the introduction, the authors described the concept of Clinical Competencies and Clinical Reasoning. However, there was a lack of statement problems regarding Clinical Competencies and Clinical Reasoning in the global, Italy, and the specific study location. More, there was also a lack of scientific reason why the study must be conducted in the Italy? What makes the difference with other countries since the issue of Clinical Competencies and Clinical Reasoning might common problems globally?
Method
1. The population of the study is registered nurses. However, this study needs clear information on the inclusion and exclusion criteria of the study participants. It can lead to selection bias. For example, what is the degree of the participant's education? What type of hospital that was included in the study? Which room? And etc..
2. Who collects the data? No information about it.
3. No sample size calculation. How do the authors validate the internal validity without sample size calculation?
4. The statement “a total of 500 questionnaires were distributed to the participating hospitals” was not clear. What does this statement mean? The total sample size?
5. Again, when was the study ended?
6. In the method section, the author did not clearly define all outcomes (main outcomes and secondary outcomes) and potential confounders. Include the potential confounders in the analysis will increase the internal validity.
7. The information of “The answers for each competence area were converted into scores between 1 and 100, where 100 represented the highest competency and 1 the lowest, in keeping with the instructions provided in the I-NCS user manual. In a similar vein, the answers for every clinical reasoning domain were recalculated using a scale that went from 15 to 75, where 75 denoted advanced clinical reasoning and 15 the lowest level.” Must be in the instrument section.
8. The authors mentioned, “To compare means between groups, inferential statistics, such as one-way analysis of variance (ANOVA), were used.” However, there is no information regarding the participant’s group. Moreover, please make the aim clearer regarding this analysis.
9. What does “process-based sufficiency of job skills” mean in the statement of “The links between the I-NCS factor scores were examined using Pearson’s correlation coefficient in order to assess the process-based sufficiency of job skills”.
10. The statement of “Calculations were made for descriptive statistics, including means and standard deviations (SD), frequency, and percentage” was also not clear. Which data were presented in means and SD. More, which data were provided in frequency and percentage?
Result
1. The statement “For this study the inclusion criterion was to be a registered nurse.” Must be in the method section.
2. The abbreviation information for each table is not complete.
3. Overall, the result tables must be reorganized to avoid misunderstanding when the reader interprets the result. Or it can lead to report bias. I found it difficult to identify the result of the correlation between Clinical Competencies and Clinical Reasoning among Italian Registered Nurses (as mentioned in the title). The result section provides the Association between socio-demographic data and the IRCS scale and the Association between socio-demographic data and the INCS scale.
Discussion and Conclusion
Overall need the “inline” issue from the title to conclusion
Author Response
REVIEWER ANSWERS 3
Title: No information about the location of the study. Study location information will help the reader identify the study easily.
- We thank the reviewer for the suggestion, in the title it is inserted that it is an Italian studio.
Abstract
- There is a duplicate word that is "ability" in the early abstract section. Please re-check the overall sentence in the manuscript to avoid typos.
- I thank the reviewer for the suggestion given we replaced the word with skill, it's a typo.
- Please emphasize the novelty of the study in the abstract section.
- We thank the reviewer for the suggestion given, we have included in the abstract a sentence that emphasizes the novelty of our study.
- What does “The data was collected from 2022.” mean? Would it have the end of the data collection period? Please make it clear.
- We thank the reviewer for the suggestion given, we have specified the data collection period, making the abstract more readable.
- Please add the statistical values of the study result.
• We thank the reviewer for the suggestion given, we have included a sentence in the abstract on the significance of the results.
Introduction
- There are some inconsistent reference styles (numbered and by name and year).
- We thank the reviewer for the suggestion given and we have corrected inconsistent reference styles.
- In the introduction, the authors described the concept of Clinical Competencies and Clinical Reasoning. However, there was a lack of statement problems regarding Clinical Competencies and Clinical Reasoning in the global, Italy, and the specific study location. More, there was also a lack of scientific reason why the study must be conducted in the Italy? What makes the difference with other countries since the issue of Clinical Competencies and Clinical Reasoning might common problems globally?
- We thank the reviewer for the suggestion, we have included a sentence that clarifies what the reviewer asked.
Method
- The population of the study is registered nurses. However, this study needs clear information on the inclusion and exclusion criteria of the study participants. It can lead to selection bias. For example, what is the degree of the participant's education? What type of hospital that was included in the study? Which room? And etc..
- We thank the reviewer for the suggestion, we have inserted a paragraph that defines the reference population, and clear information on the inclusion and exclusion criteria
- Who collects the data? No information about it.
- We thank the reviewer for the suggestion given, we have included some words that define better who has collected the data.
- No sample size calculation. How do the authors validate the internal validity without sample size calculation?
- We thank the reviewer for the suggestion, we have inserted a sentence describing the sample size calculation
- The statement “a total of 500 questionnaires were distributed to the participating hospitals” was not clear. What does this statement mean? The total sample size?
- We thank the reviewer for the suggestion, yes the questionnaires were distributed with the intention of reaching the sample size we had set ourselves.
- Again, when was the study ended?
- We thank the reviewer for the suggestion, in the design section of the study we clearly stated the date when the study finished 5 march 2022
- In the method section, the author did not clearly define all outcomes (main outcomes and secondary outcomes) and potential confounders. Include the potential confounders in the analysis will increase the internal validity.
-
Repeat the main outcome objective and secondary outcomes sub-objectives
- The information of “The answers for each competence area were converted into scores between 1 and 100, where 100 represented the highest competency and 1 the lowest, in keeping with the instructions provided in the I-NCS user manual. In a similar vein, the answers for every clinical reasoning domain were recalculated using a scale that went from 15 to 75, where 75 denoted advanced clinical reasoning and 15 the lowest level.” Must be in the instrument section
- We thank the reviewer for the suggestion, the calculation of the scores of the scales used has been included in the tools section.
- The authors mentioned, “To compare means between groups, inferential statistics, such as one-way analysis of variance (ANOVA), were used.” However, there is no information regarding the participant’s group. Moreover, please make the aim clearer regarding this analysis.
- We thank the reviewer for the suggestion we made regarding the use of ANOVA in our study
- What does “process-based sufficiency of job skills” mean in the statement of “The links between the I-NCS factor scores were examined using Pearson’s correlation coefficient in order to assess the process-based sufficiency of job skills”.
- We thank the reviewer for his suggestion: we have modified the sentence to improve its context and make it more readable.
- The statement of “Calculations were made for descriptive statistics, including means and standard deviations (SD), frequency, and percentage” was also not clear. Which data were presented in means and SD. More, which data were provided in frequency and percentage?
- We thank the reviewer for the suggestion given, we have improved and changed the sentence making it more readable in the interpretation of the data.
Result
- The statement “For this study the inclusion criterion was to be a registered nurse.” Must be in the method section.
- We thank the reviewer for the suggestion, we have removed the sentence from the Results section and placed it in the method section.
- The abbreviation information for each table is not complete.
- We tried to eliminate many abbreviations that did not allow a full comprehension of the tables.
- Overall, the result tables must be reorganized to avoid misunderstanding when the reader interprets the result. Or it can lead to report bias. I found it difficult to identify the result of the correlation between Clinical Competencies and Clinical Reasoning among Italian Registered Nurses (as mentioned in the title). The result section provides the Association between socio-demographic data and the IRCS scale and the Association between socio-demographic data and the INCS scale.
- We thank the reviewer for the suggestion given, we have included some changes in the data tables for better interpretation, we have also included a table that was missing on correlations, we apologize to the reviewer.
Discussion and Conclusion
Overall need the “inline” issue from the title to conclusion
- We modified the discussion and conclusions, and we included sentences to be in line with the meaning and the purpose of the study.
Reviewer 4 Report
Comments and Suggestions for Authors
Thank you for addressing an important issue in nursing (competencies and reasoning). The manuscript written and organized in a good way, however, few minor notes should be addressed and modified before moving to the next step in publication.
Abstract
1) In the first line, why repeating the word (ability)?
2) More results are needed in the abstract regarding levels and associations of competencies and reasoning.
3) Modify the conclusion to reflect your study results.
Introduction
4) In paragraph 5, what is the reason for repeating (skills) twice?
5) in paragraph 11; 2010 is not a recent study.
Study design
6) clarify what do you mean by (observational)? I think a cross-sectional is enough.
7) In the second paragraph; mention what do you mean by two questionnaires (I guess NCS and NCRS)?
Instrument
8) Clarify what are the instructions at the beginning of each instrument? For example, how nurses perceive their competencies of reasoning, how they evaluate their own competencies and reasoning?
9) Any cut-off points for NCRS?
Statistical Analysis
10) I did not see any use for ANOVA.
Results
11) Data collection: 500 questionnaires were administered or distributed?
12) Table 1, put (Role) under title.
13) Table 1, have the title in the first column, and the categories in the second one (Just replace them).
14) You are using IRCS starting from page 6. This confused the reader if it the same of NCRS. You need to be consistent in using the tools names.
15) Table 3 and 4. They are too long with unneeded data. I suggest only to have the title of each demographic (seven titles) and only F and significance level.
16) You have been using DS, Do you mean SD (standard Deviation)?
17) in Tables 4; Categories of Job satisfaction (enough, A lot, by no means, little) are not consistent with those categories in table 1 (Quite, A lot, For nothing, and a Little).
18) report the result of the levels of competencies and reasoning at the beginning of the result section.
Discussion
19) Paragraph two, how did you come to a conclusion of (excellent) average for the two measures. Did you relay on the cut-off points?
20) Paragraph two, report the average of competencies and reasoning (77.46 and 62.98, respectively).
21) In paragraph 6, you indicate to refer to table 5. There are no table 5.
22) As you discuss the results of competencies and reasoning with job satisfaction. Please do the same for Role, Age, and Sex with INCS; and operating unit with IRCS.
Conclusion
23) the first line, how reasoning have no impact on competencies. Clarify how did you reach to such a conclusion.
Reference
24) be consistent in using the Journal format. For example some times you used full name of the Journals and some times you used the abbreviated ones.
Thank you
.
Author Response
REVIEWER ANSWERS 4
Thank you for addressing an important issue in nursing (competencies and reasoning). The manuscript written and organized in a good way, however, few minor notes should be addressed and modified before moving to the next step in publication.
Abstract
1) In the first line, why repeating the word (ability)?
• I thank the reviewer for the suggestion we replaced the word with skill, it's a typo.
2) More results are needed in the abstract regarding levels and associations of competencies and reasoning.
• We thank the reviewer for the suggestion given: we have included a sentence within the abstract that better defines the levels of significance between the two scales used within our study
3. Modify the conclusion to reflect your study results.
• We thank the reviewer for the suggestion given, we have modified the text making it more readable and congruous with the conclusions of our study.
Introduction
4) In paragraph 5, what is the reason for repeating (skills) twice?
• I thank the reviewer for the suggestion we replaced the word with "ability", it's a typo.
5) in paragraph 11; 2010 is not a recent study.
• We thank the reviewer for the suggestion given, we have removed the word "recent", as it could give a wrong interpretation.
Study design
6) clarify what do you mean by (observational)? I think a cross-sectional is enough.
• We thank the reviewer for the suggestion given, we have removed the word cross-sectional as recommended.
7) In the second paragraph; mention what do you mean by two questionnaires (I guess NCS and NCRS)?
• We thank the reviewer for the suggestion given, we have better defined the acronym of the scales used, I make the reading better.
Instrument
8) Clarify what are the instructions at the beginning of each instrument? For example, how nurses perceive their competencies of reasoning, how they evaluate their own competencies and reasoning?
• We thank the reviewer for his suggestion and we have included a few words in the period describing what both the INCS and NCRS scale rate.
9) Any cut-off points for NCRS?
• We thank the reviewer for the suggestion we have included the cut-off for the NCRS scale
Statistical Analysis
10) I did not see any use for ANOVA.
• We thank the reviewer for the suggestion given, we have included a sentence that improves the readability of the sentence.
Results
11) Data collection: 500 questionnaires were administered or distributed?
• We thank the reviewer for the suggestion given, we changed the word administered to distributed
12) Table 1, put (Role) under title.
• We thank the reviewer for the suggestion we have replaced the word TITLE with ROLE, as indicated by the reviewer.
13) Table 1, have the title in the first column, and the categories in the second one (Just replace them).
• We thank the reviewer for the suggestion given, we made the changes in table 1 as recommended by the reviewer, now the reading of the table is improved
14) You are using IRCS starting from page 6. This confused the reader if it the same of NCRS. You need to be consistent in using the tools names.
• We thank the reviewer for the suggestion given, we have changed IRCS to I-NCRS as recommended by the reviewer
15) Table 3 and 4. They are too long with unneeded data. I suggest only to have the title of each demographic (seven titles) and only F and significance level.
• We thank the reviewer for the suggestion, we have modified the tables according to the information given, now they are clearer for the reader.
16) You have been using DS, Do you mean SD (standard Deviation)?
• We thank the reviewer for the suggestion, yes it was a typo, we apologize for the error.
17) in Tables 4; Categories of Job satisfaction (enough, A lot, by no means, little) are not consistent with those categories in table 1 (Quite, A lot, For nothing, and a Little).
• We thank the reviewer for the suggestion, we have changed the table now the Job Satisfaction categories are not present, as the reviewer had indicated above.
18) report the result of the levels of competencies and reasoning at the beginning of the result section.
• We thank the reviewer for the suggestion the requested results are in the middle in the results section
Discussion
19) Paragraph two, how did you come to a conclusion of (excellent) average for the two measures. Did you relay on the cut-off points?
• We thank the reviewer for the suggestion, yes we have come to these conclusions based on the cut-offs, we have included a paragraph in the tools section that explains the cut-offs better.
20) Paragraph two, report the average of competencies and reasoning (77.46 and 62.98, respectively).
• We thank the reviewer for the suggestion, we have edited the sentence as indicated by the reviewer.
21) In paragraph 6, you indicate to refer to table 5. There are no table 5.
• We thank the auditor for his suggestion: we have included Table 5 with reference to the correlations between the two scales, INCS and NCRS
22) As you discuss the results of competencies and reasoning with job satisfaction. Please do the same for Role, Age, and Sex with INCS; and operating unit with IRCS.
• We thank the reviewer for the suggestion given, we have included a sentence about it.
Conclusion
23) the first line, how reasoning have no impact on competencies. Clarify how did you reach to such a conclusion.
• We thank the reviewer for the suggestion, we have included a sentence to clarify how we arrived at these conclusions.
Reference
24) be consistent in using the Journal format. For example some times you used full name of the Journals and some times you used the abbreviated ones.
• We thank the reviewer for the suggestion given, we have fixed the journals in the bibliography with the medline abbreviations
Thank you
Round 2
Reviewer 1 Report
Comments and Suggestions for Authors
When I submitted my initial review, I expressed concern that the literature was too old and I see that the authors have refreshed somewhat but not within the last 5 years. Out of 40 references, only 3 are 5 years or newer. On a quick search through CINAHL, using nursing and competency I found over 3,000 references for peer-reviewed articles from 2020 to 2024. The introduction is not substantial and does not lay an accurate or scholarly foundation for the topic. The language used is simplistic and not informative.
The conclusion described in the abstract does not match the conclusion stated in the article “ The results of our research revealed that nurses apply their competencies to every care situation, through clinical reasoning, using a set of basic and advanced knowledge.” It is not a new concept, nor is it newsworthy. The final statement of the abstract reads “This study represents the first attempt to analyze the relationship between clinical competencies and clinical reasoning.” I searched CINAHL using the terms clinical competence and clinical reasoning and found 342 articles in this database alone describing this relationship.
The conclusion cannot be derived from the data or the text. . "We can therefore deduce that clinical reasoning does not influence the development of clinical competencies and, consequently, does not adequately prepare a nurse for the various contexts of clinical practice." There is no way this statement could be deduced from the data shown. It is contradictory to this statement earlier in the paper "The results of our study indicate that the clinical reasoning and clinical competencies of the examined nurses seem to exhibit ongoing progress. This is supported by statistically significant correlations between the various variables examined and the scores of the dif different dimensions of the I-NCS and I-NCRS scales (Tables 3 and 4)."
Comments on the Quality of English LanguageOnly one grammatical issue, in line 318 the word "teached" was used instead of "taught."
Author Response
Comment 1: When I submitted my initial review, I expressed concern that the literature was too old and I see that the authors have refreshed somewhat but not within the last 5 years. Out of 40 references, only 3 are 5 years or newer. On a quick search through CINAHL, using nursing and competency I found over 3,000 references for peer-reviewed articles from 2020 to 2024. The introduction is not substantial and does not lay an accurate or scholarly foundation for the topic. The language used is simplistic and not informative.
Response 1: We thank the reviewer for the suggestion given, we have revised the introduction also adding an updated bibliography that is more congruent with clinical expertise and clinical reasoning.
Comment 2: The conclusion described in the abstract does not match the conclusion stated in the article “The results of our research revealed that nurses apply their competencies to every care situation, through clinical reasoning, using a set of basic and advanced knowledge.” It is not a new concept, nor is it newsworthy. The final statement of the abstract reads “This study represents the first attempt to analyze the relationship between clinical competencies and clinical reasoning.” I searched CINAHL using the terms clinical competence and clinical reasoning and found 342 articles in this database alone describing this relationship.
Response 2: We have deleted the sentence "The results of our research revealed that nurses apply their competencies to every care situation, through clinical reasoning, using a set of basic and advanced knowledge." Because it could be confusing, and we put in a new sentence for what this study represents.
Comment 3: The conclusion cannot be derived from the data or the text. . "We can therefore deduce that clinical reasoning does not influence the development of clinical competencies and, consequently, does not adequately prepare a nurse for the various contexts of clinical practice." There is no way this statement could be deduced from the data shown. It is contradictory to this statement earlier in the paper "The results of our study indicate that the clinical reasoning and clinical competencies of the examined nurses seem to exhibit ongoing progress. This is supported by statistically significant correlations between the various variables examined and the scores of the dif different dimensions of the I-NCS and I-NCRS scales (Tables 3 and 4)."
Response 3: We thank the reviewer for his suggestion, we have modified the first sentence to make it understandable, while we have removed the second sentence because it was confusing.
Reviewer 3 Report
Comments and Suggestions for Authors
Abstract
1. Please emphasize the novelty of the study in the abstract section.
2. Please add the statistical values of the study result.
Method
1. The population of the study is registered nurses. However, this study needs clear information on the inclusion and exclusion criteria of the study participants. It can lead to selection bias. For example, what is the degree of the participant's education? What type of hospital that was included in the study? Which room? And etc..
2. Who collects the data? No information about it. Researcher?
Author Response
Abstract
Comment 1: Please emphasize the novelty of the study in the abstract section.
Response 1: We thank you for the reviewer's suggestion, we have inserted a sentence that emphasizes the novelty of our study
Comment 2: Please add the statistical values of the study result.
Response 2: We thank the reviewer for the suggestion given: we have included statistical data in the abstract
Method
Comment 3: The population of the study is registered nurses. However, this study needs clear information on the inclusion and exclusion criteria of the study participants. It can lead to selection bias. For example, what is the degree of the participant's education? What type of hospital that was included in the study? Which room? And etc..
Response3: We thank the reviewer for the suggestions given, we have inserted some sentences in the paragraph of the study design that specify what the reviewer has suggested
Comment 4: Who collects the data? No information about it. Researcher?
Response 4: We thank the reviewer for the suggestion given, we have included a sentence and the initials of the researcher who managed the data